# ICFI: A Feature Importance Measure for Multi-Class Classification

## Abstract

Feature importance is one of the most prominent methods in explainable artificial intelligence. It seeks to score the features an artificial intelligence model relies on the most. In multi-class classification, current methods fail to explain inter-class relationships as they either provide explanations for binary classification only, or suffer from aggregation bias. In a multi-class classification scenario, features may carry discriminative power to separate some of the classes while being otherwise less relevant. State-of-the-art feature importance measures do not capture this behavior. We propose Inter-Class Feature Importance (ICFI), a measure that scores the feature importance to discriminate between an arbitrary pair of classes. ICFI is a post-hoc, model-agnostic method, independent from the machine learning architecture employed. ICFI marginalises the target output with respect to the feature of interest, leveraging the resulting change in model behavior to quantify feature importance. We present ICFI's properties and argue its relevance, describing use cases and showing insights gained. We demonstrate through thorough experiments on real-world datasets how ICFI captures the features characteristics for specific class relationships.

## 1 Introduction

The eXplainable Artificial Intelligence (XAI) research field focuses on making Machine Learning (ML) models understandable to human stakeholders (König et al., 2021). ML models' increasing complexity is proving a hurdle in complying with legal requirements (König et al., 2021; European Parliament & Council of the European Union; Tritscher et al., 2023), validating architectures in high-stake scenarios (Dinu et al., 2020), and treating protected groups fairly (Caton & Haas, 2024). XAI research aims to solve these problems. Specifically, Feature Importance (FI) is one of the most popular XAI methods (Saarela & Jauhiainen, 2021). It quantifies the relevance of each input feature for the model prediction (Muschalik et al., 2023), allowing verification of whether the importance of the features aligns with background knowledge (Alfeo et al., 2023).

If an XAI method works with a trained ML model, it is classified as post-hoc (Tritscher et al., 2023). Post-hoc techniques are flexible and can be applied to existing models to improve interpretability (Das & Rad, 2020). Additionally, a FI method that applies to any ML model is considered model-agnostic (Tritscher et al., 2023) and does not impose constraints on the model architecture.

Current FI methods either provide explanations for single instances (locally) in a binary classification setting, quantifying importance toward the positive class, or explain the whole dataset (globally). These two types of explanation cannot capture inter-class relationships. Local methods are targeted toward binary settings. For global methods, the importance ranking stands valid for a certain percentage of the population, but it may not be accurate for all of the population; this effect is known as aggregation bias (Mehrabi et al., 2022).

A concrete example is the kidney cancer recognition case (Muhamed Ali et al., 2018; Cancer Genome Atlas Research Network et al., 2013), where clinical data and RNA sequencing information are utilized to detect cancer sub-types. Such a case is a multi-class classification task. After building a classifier, it is possible to obtain insight into the model behavior by applying XAI methods and obtaining the global FI. The output is a single ranking, quantifying feature contributions. This single ranking, however, does not consider class relationships. Certain features might be important to separate two specific cancer sub-types while being otherwise less relevant. Existing FI methods fail

to capture this. Moreover, a feature crucial for differentiating between two cancer types may not be found globally important, which risks critical oversights.

We can use the traditional Confusion Matrix (CM) to evaluate the model, quantifying pairwise class combinations' false positives and negatives (Beauxis-Aussalet & Hardman, 2014). We might notice that the model does not distinguish between classes equally well, notable in the CM entries. How do we understand why this happens? We know that one CM entry describes the model's misclassifications between two classes. We need the FI importance (the explanations) for when the model separates the classes to know why it misclassifies them. If we use global methods, we will receive an aggregation of every sample regardless of their class, which fails to provide a specific explanation for the misclassification between two classes.

To address this gap, we look towards a method that outputs a feature ranking for any pair of classes, quantifying the importance of features in separating the two classes. Hence, we propose a new feature importance measure to explain an ML model in a multi-classification scenario: Inter-Class Feature Importance (ICFI). We seek to identify the features the model relies on the most when trying to discriminate between two classes. Applying existing FI measures to the classes of interest would not achieve our goal. We would still evaluate the importance of the overall classification task, not capturing relationships between the two classes. When we have a binary classification task, separating the two classes coincides with the overall model's task, ICFI is thus a generalization of binary FI.

ICFI, to compute FI, permutes the feature of interest to mimic the absence of the feature from the model as similarly done in Permutation Feature Importance (PFI) (Fumagalli et al., 2023; Strobl et al., 2008) and other XAI methods Fisher et al. (2019). We quantify feature importance by evaluating model performance with and without the information carried by the feature inspected. Permutation allows our measure to be completely model-agnostic without needing to train any additional model. ICFI is a model-agnostic and post-hoc method, and its simple algorithmic implementation encourages its use in any scenario involving tabular data.

Our core contribution includes an introduction of ICFI, a model-agnostic XAI method for quantifying feature importance in separating an arbitrary pair of classes. An overview of ICFI's properties and an in-depth discussion of its relevance providing use cases. Through empirical evaluation of real-world datasets, we show how ICFI offers new insights into the inner workings of an ML model.

The structure of the remainder of the paper firstly introduces related work in Section 2, then ICFI is introduced in Section 3. Penultimately, we explore experiments in Section 4 and Section 5 concludes the paper discussing possible future work.

## 2 RELATED WORK

The XAI research field is very dynamic, with recent surveys providing detailed overviews (Ali et al., 2023; Das & Rad, 2020; Theissler et al., 2022). XAI literature addresses both model-agnostic and model-specific approaches as well as post-hoc explanations. Model-specific approaches are tailored for a specific model or class of models only (Sundararajan et al., 2017; Carletti et al., 2023; Bach et al., 2015; de Sá, 2019). Post-hoc methods target fully trained models (Ali et al., 2023).

In our work, we focus on feature importance methods. FI methods generate explanations by pointing out the model's most important features (Das & Rad, 2020).

The most prominent and widely used methods in this domain include SHapley Additive exPlanations (SHAP) (Lundberg & Lee, 2017) and Local Interpretable Model-agnostic Explanations (LIME) (Ribeiro et al., 2016). SHAP applies Shapley Values from game theory to locally assess each feature's importance in machine learning (Ali et al., 2023). SHAP provides a unique feature ranking only in binary classification. Global SHAP explanations can be retrieved by averaging local ones. They do not take into account inter-class relationships and they misrepresent samples whose importance does not align with the average, suffering from aggregation bias (Mehrabi et al., 2022). SHAP's usage in diverse research areas (Cooper et al., 2021; Antwarg et al., 2021; García & Aznarte, 2020) underscores its effectiveness while it's high computational complexity makes its implementation challenging (Muschalik et al., 2023).

On the other hand, LIME is a local XAI method that quantifies feature importance by approximating the model locally with an inherently interpretable surrogate, responsible to provide the explanation (Adamczewski et al., 2020; Ribeiro et al., 2016). Global feature rankings can be retrieved through aggregating local instances (Ribeiro et al., 2016). LIME may exhibit instability if slight changes in the surrogate's input occur (Zhou et al., 2021).

Global XAI approches include Partial Dependence Plots (PDP) (Friedman, 1991) and Permutation Feature Importance (Breiman, 2001). PDPs are a low-dimensional graphical representation showing the dependence between the target and a set of features of interest (Greenwell et al., 2017). PDPs do not target feature importance rankings but aim to visualize the interaction between the target variable and a set of input features.

PFI, introduced in Breiman (2001) for random forests, assesses change in the model's performance when permuting the feature of interest, effectively marginalizing the other features (Fumagalli et al., 2023; Strobl et al., 2008). Permutation, aims at mimicking the absence of the feature of interest. If a feature provides information about the target variable, breaking the association through permutation will be reflected in the model's performance (Strobl et al., 2008). The feature is deemed unimportant when there is no significant increase in the empirical risk after permuting (Debeer & Strobl, 2020). A slight decrease in risk is also possible and is attributed to chance or to a sub-optimal model (Debeer & Strobl, 2020; Fisher et al., 2019).

PFI has been the focus of several research articles addressing both applications and enhancements (König et al., 2021; Wang et al., 2016; Strobl & Zeileis, 2008; Strobl et al., 2008; Nicodemus et al., 2010; Epifanio, 2017). Specifically, Fisher et al. (2019) expand PFI making it model-agnostic and not limited to trees or ensemble models (Fumagalli et al., 2023).

## 3 FEATURE IMPORTANCE IN MULTI-CLASS CLASSIFICATION

We propose ICFI, with the aim of providing feature importance for the task of separating a pair of classes in a multi-class classification scenario. Some features may indeed carry discriminative power to discriminate two classes only, while they might not be otherwise leveraged by the model. Hence, they might not be highlighted as important by global methods.

We keep a running simple example, using the Iris dataset (Fisher, 1936), to gain intuition for our method. The Iris dataset is a popular benchmark dataset for multi-class classification tasks. It consists of 150 samples of iris flowers belonging to three separate species: *versicolor*, *virginica* and *setosa*. The objective is to classify flowers' samples leveraging four features, describing respectively *sepal length*, *sepal width*, *petal length* and *petal width*.

Here, we use only *sepal length* and *sepal width* for sake of example and simplicity (Figure 1). Describing the Iris dataset through *sepal width* and *sepal length* causes the *versicolor* and *virginica* classes to overlap (Zaki & Meira, 2014). As also testified by Figure 1, *setosa* is linearly separable from the other two classes. *Versicolor* and *virginica* instead, overlap with each other in the two dimensional input space. The expectation is that a classifier will struggle to separate between *versicolor* and *virginica*.

We fit a decision tree classifier on the simplified Iris dataset, using $50\%$ of the data for training. Figure 1 shows the decision boundary. As expected, several *virginica* samples are misclassified as *versicolor* and multiple *versicolor* flowers are wrongly labeled as *virginica*. This testifies how the model does not effectively separate *versicolor* and *virginica*. On the contrary, no *setosa* sample is classified as *virginica* while just one is wrongly assigned to the *versicolor* class.

To capture feature importance to discriminate two classes, we need to describe how the model performs when separating the pair of classes. Referrnewsgrouping to Figure 1, we need to quantify the model's inefficiency in separating *virginica* and *versicolor*.

Global FI methods, don't look into inter-class relationships and suffer from aggregation bias. This work aims to evaluate performance in the task of discriminating two classes as a pathway to Inter-Class Feature Importance.

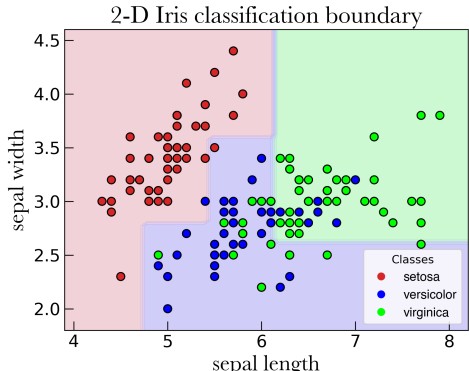

Figure 1: Multi-class classification example. The Iris dataset projected on the *sepal length* and *sepal width* features is used to fit a decision tree. The 3 classes' points are coloured in blue, green and red for *versicolor*, *virginica* and *setosa* respectively. The tree's decision boundary is indicated by the different background colors.

### 3.1 INTER-CLASS FEATURE IMPORTANCE

ICFI takes into consideration two classes and outputs the feature importance for the model task of separating them. As in binary FI methods, we do not change the computation based on the order of the classes. Hence, we do not distinguish the importance of a feature in separating class $\sigma$ with class $\rho$ from the importance in separating class $\rho$ with class $\sigma$. This requires ICFI to be symmetrical with respect to the two classes of interest. When a feature is deemed unimportant, we quantify its importance with $0$ as intuition would suggest and relating to the discussion on missingness, in Lundberg & Lee (2017). Missingness is the property stating that features missing from the original input get an attribution of $0$. Furthermore, humans reason better when dealing with a small limited range than with a potentially infinitely high value (Resnick et al., 2017; Jones et al., 2008). An unbounded number for the importance quantification is a feature detrimental to interpretability, as a FI value with no reference hinders its interpretation (Adamczewski et al., 2020; Pries et al., 2023). We thus want our measure to be also upper bounded. Next, we introduce the formalism used in this paper.

Given a domain set $\mathcal{X}$, a label set $\mathcal{Y}$, a classifier $h : \mathcal{X} \to \mathcal{Y}$ and a loss function $l$, we define the true risk $R_t(h)$ as the expected loss of $h$ with respect to a probability distribution $\mathcal{D}$ over $\mathcal{X} \times \mathcal{Y}$ (Shalev-Shwartz & Ben-David, 2014),

$$\mathcal{R}_t(h) = \mathbf{E}_{z \sim D}\left[l(h, z)\right] \quad . \tag{1}$$

For convenience, we drop the $h$ in the risk's notation. $\tilde{\mathcal{R}}_{tj}$ denotes the true risk after permuting feature $j$.

In order to evaluate our measure, instead of the true risk which is not computable as the ML model has no access to $\mathcal{D}$ (Shalev-Shwartz & Ben-David, 2014), we approximate it using the empirical risk $R(h)$, i.e., the average loss over a given data sample $(z_1, ..., z_N)$ (Shalev-Shwartz & Ben-David, 2014), defined as: $R(h) = \frac{1}{N} \sum_{i=1}^{N} l(h, z_i)$.

We permute a feature by uniformly sampling one of its possible permutations. This means that if our data sample consists of $N$ records, each of the $N!$ permutations can be selected with probability $\frac{1}{N!}$.

The intuition behind ICFI hinges on the fact that the more misclassifications between classes $\sigma$ and $\rho$, the more the empirical risk will decrease when we merge the two classes. Merging a pair of classes means they are now considered one class, where all misclassifications between the pair will become successful classifications, effectively improving model performance. We compute the decrease in empirical error when merging two classes through

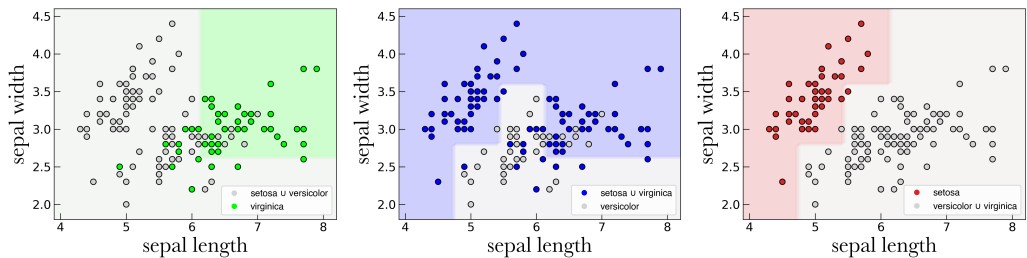

Figure 2: Multi-class classification example. From left to right we merged classes $(0 + 1)$, $(0 + 2)$, $(1 + 2)$. Points belonging to merged classes are coloured in gray. Only the decision boundary between the two classes is pictured.

$$\Delta R^{\sigma \rho} = \mathcal{R} - \mathcal{R}^{\sigma \rho} \quad , \tag{2}$$

with $\mathcal{R}^{\sigma \rho}$ the empirical error when merging classes $\sigma$ and $\rho$. In Eq. 2, as in the remainder of the paper, we use Greek letters to refer to classes while we refer to features with Latin letters.

Eq. 2 would be incomplete without a clearer definition of the merging operation. We deal with two cases: models which ouput class probabilities and model that output the predicted class only. Considering the case in which probabilities are provided, when merging, two classes are considered as one, resulting in a combined probability through summing. The merging definition can be adapted when models do not provide class probabilities. In this scenario, merging class $\sigma$ with class $\rho$ would mean changing model outputs from $\sigma$ to $\rho$. Regardless of the model, during evaluation, $\sigma$-labelled target data points are labelled as $\rho$. Merging is symmetric, i.e., merging $\sigma$ with $\rho$ is equivalent to merging $\rho$ with $\sigma$, making ICFI symmetric as desired.

If a model struggles to separate two classes, Eq. 2 will reflect it. To probe ICFI's intuition of relying on the empirical risk decreasing when merging two classes, we look back at the example displayed in Figure 1. For the sake of the example, we use a mathematically simple loss function: the *zero-one* loss. The *zero-one* loss outputs 1 for a misclassified sample and 0 otherwise.

Dealing with three classes, three pairwise merges are considered, using all available samples. We merge *versicolor* with *virginica*, *setosa* with *versicolor* and *setosa* with *virginica*. The three scenarios are depicted in Figure 2 where points belonging to merged classes are in gray and only the decision boundary between the two resulting classes is visible. The computed decreases in empirical error $\Delta R^{\sigma \rho}$ are respectively 0.20, 0.01 and 0.

As the decision boundary in Figure 1 indicates, the model struggles the most separating *versicolor* and *virginica*. $\Delta R^{\sigma \rho}$ reflects this, taking the highest value when merging *versicolor* with *virginica*. The model making no misclassifications between *setosa* and *virginica* is underscored by the null decrease in empirical error. Lastly, the only two misclassifications between *setosa* and *versicolor* cause a low 0.01 value of $\Delta R^{\sigma \rho}$.

The example in Figure 1, thus shows how the decrease in accuracy $\Delta R^{\sigma \rho}$ captures model performance in separating classes. This makes $\Delta R^{\sigma \rho}$ a key ICFI component.

To evaluate feature importance for feature $j$, we permute $j$ and measure the difference in model performance. To evaluate ICFI we thus compute $\Delta R^{\sigma \rho}$ when permuting feature $j$ measuring

$$\Delta \tilde{\mathcal{R}}_j^{\sigma \rho} = \tilde{\mathcal{R}}_j - \tilde{\mathcal{R}}_j^{\sigma \rho} \quad , \tag{3}$$

with $\tilde{\mathcal{R}}_j^{\sigma \rho}$ being the empirical error when permuting feature $j$, and merging classes $\sigma$ and $\rho$.

Eq. 2 and Eq. 3 evaluate decrease in empirical error respectively before and after permuting. ICFI could thus be straightforwardly defined taking the difference $\Delta \tilde{\mathcal{R}}_j^{\sigma \rho} - \Delta R^{\sigma \rho}$ or through the ratio $\frac{\Delta \tilde{\mathcal{R}}_j^{\sigma \rho}}{\Delta R^{\sigma \rho}}$. Both functional forms result in an unbounded target range, which goes against our specified requirements for interpretability. Additionally, setting an upper bound for the feature importance

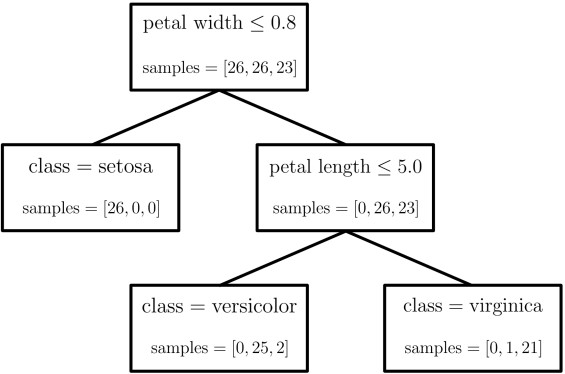

Figure 3: Decision tree of depth 2 classifying the Iris dataset, one of the most popular multi-class classification benchmark datasets. The *samples* entry indicate how many training samples for class *setosa*, *versicolor* and *virginica* respectively, go in a specific node.

quantification, allows us to assess whether a feature is highly important according to ICFI's definition. Without a limit on the importance evaluation, only assessments relative to other computed importances could be made.

We can get a bounded measure by constraining the target range between $0$ and $1$, with $0$ signaling a non-important feature. Starting from the ratio $\frac{\Delta \tilde{\mathcal{R}}_j^{\sigma\rho}}{\Delta R^{\sigma\rho}}$, which has the advantage of normalizing the measure by the model performance before permutation, we need a function $f(x)$ mapping the output interval $[1, +\infty)$ to $[0, 1]$. $f$ should also be strictly increasing in order to preserve feature ranking. Choosing $f$ as: $f(x) = 1 - \frac{1}{x}$, would lead to defining $ICFI_j^{\sigma\rho}$ as $1 - \frac{\Delta R^{\sigma\rho}}{\Delta \tilde{\mathcal{R}}_j^{\sigma\rho}}$. The problem with this definition is that actually, while $\tilde{R}_j$ is smaller than $R$ only in rare instances dictated by chance or by a sub-optimal model (Debeer & Strobl, 2020; Fisher et al., 2019), $\Delta \tilde{\mathcal{R}}_j^{\sigma\rho}$ can be smaller than $\Delta R^{\sigma\rho}$ because the model distinguishes better the two classes after permutation. This would lead $1 - \frac{\Delta R^{\sigma\rho}}{\Delta \tilde{\mathcal{R}}_j^{\sigma\rho}}$ to be negative. To see that $\Delta \tilde{\mathcal{R}}_j^{\sigma\rho}$ can be smaller than $\Delta R^{\sigma\rho}$, we look at an application example.

Figure 3 shows a decision tree classifier fitted on the Iris dataset (Fisher, 1936) with all four features. The *samples* field shows how many training samples for class *setosa*, *versicolor* and *virginica*, respectively, go in a specific node. The model perfectly classifies the *setosa* class, and the *setosa* leaf does not include *versicolor* or *virginica* class samples. Thus, in the setting where we do not permute any of the features, when the model deals with samples belonging to the *versicolor* and *virginica* class, it will typically assign them to the *versicolor* or *virginica* leaves. Leaves where the model can misclassify the two classes, as showed by the *samples* field. For example, we can see from the *versicolor* leaf, that 25 training samples belonging to the *versicolor* class are correctly labeled as *versicolor* while 2 samples are misclassified as *virginica*. Conversely, in the *setosa* leaf, no *versicolor* sample is misclassified as *virginica* and vice-versa.

Consider losing the information carried by the *petal width* feature by permuting it. The model will likely send more *virginica* and *versicolor* samples to the left branch, as we loose the discriminative power which fully separates *setosa* from the *versicolor* and *virginica* samples. In the left branch, solely composed by the *setosa* leaf, the model does not misclassify *versicolor* with *virginica*, as suggested by the *samples* field, and merging *versicolor* with *virginica* will lead to a low decrease in empirical error. Thus, in this case, we expect $\Delta \tilde{\mathcal{R}}_j^{\sigma\rho}$ to be smaller than $\Delta R^{\sigma\rho}$.

Hence, we are not interested in the difference $\Delta \tilde{\mathcal{R}}_j^{\sigma\rho} - \Delta R^{\sigma\rho}$ but in its absolute value, i.e., $|\Delta \tilde{\mathcal{R}}_j^{\sigma\rho} - \Delta R^{\sigma\rho}|$. The absolute value estimates by how much model performance differs after permutation which is what we need to quantify feature importance while having a positive measure, as outlined in the desired ICFI properties.

We thus propose to account for the above mentioned requirements by defining $ICFI_j^{\sigma\rho}$ as

$$ICFI_j^{\sigma\rho} = 1 - \frac{1}{1 + \left|\Delta\tilde{\mathcal{R}}_j^{\sigma\rho} - \Delta R^{\sigma\rho}\right|/\Delta R^{\sigma\rho}} \quad , \tag{4}$$

$ICFI_j^{\sigma\rho}$ quantifies the importance of feature $j$ in the task of separating classes $\sigma$ and $\rho$.

ICFI's formulation in Eq. 4 respects the above mentioned requirements of ICFI being non-negative, bounded and symmetric with respect to the classes inspected. We prove these claims in Appendix A.4.

## 4 EXPERIMENTS

One of XAI's biggest challenges is its evaluation, which cannot generally rely on quantifiable metrics like accuracy. Furthermore, ground-truth and evaluation standards are often lacking (Adamczewski et al., 2020; Afchar et al., 2021; Molnar et al., 2023; Pries et al., 2023; Ali et al., 2023; Tritscher et al., 2023). When we do have ground-truth on which features should be important, it usually refers to the data, not to the model. Explanations are commonly assessed either as a by-product of accuracy or through case studies in application contexts. Our experiments[1] utilize both assessment methods through real-world datasets. Different classifiers are used throughout the experiments in order to highlight the model-agnostic nature of the proposed method. In our experiments, we do not target model performance as such; this paper focuses on the quality of explanations. Unless stated otherwise, we train models which output class probabilities, allowing us to use the cross-entropy loss, widely employed in classification tasks (Zhang & Sabuncu, 2018; Mao et al., 2023) and determining how classes are merged.

In 4.1, an interpretable model is fitted to the Iris dataset (Fisher, 1936) in order to have ground-truth in our feature importance rankings. We then seek to quantify feature importance and test whether ICFI correctly retrieves ground-truth information. Section 4.2 exploits background knowledge in a NLP dataset to evaluate ICFI. In this setting, words are used as features and some words are expected to be highly discriminative for a specific pair of class while are not supposed to be leveraged by the model in other pairs. We test if ICFI correctly captures this behavior. Lastly, Our measure is compared to three different benchmarks in Section 4.3 to evaluate the quality of the feature importance rankings. Throughout the experiments, PFI is computed as a global feature importance measure for the sake of comparison. We use PFI for two reasons. First, it is a vastly employed, model-agnostic, inherently global FI method. Second, it uses a strategy similarly to the one of ICFI, i.e., it permutes the feature of interest evaluating model performance before and after permutation. ICFI and PFI are computed on test data.

### 4.1 3-CLASS DECISION TREE, NUMERICAL

The *Iris* dataset (Fisher, 1936) is a multi-class classification dataset. The task consists in distinguishing three different types of flowers i.e., *Setosa*, *Versicolor* and *Virginica*, described by four features: *petal width*, *petal length*, *sepal width* and *sepal length*. We fit a CART decision tree (Breimann et al., 1984), setting the maximum depth at two as shown in Figure 3.

The decision tree has the advantage of being an inherently interpretable model. As we already discussed, we seldom have ground-truth in the model and an interpretable one, together with a relatively simple dataset, offers the ground-truth needed to evaluate ICFI. The tree structure in Figure 3, provides a feature importance ranking. First of all, the tree leverages just two features: *petal width* and *petal length*. *Sepal length* and *sepal width* should thus be labeled as unimportant. Moreover, features leveraged close to the root have a higher global influence than the ones used in lower nodes (Laugel et al., 2018). We thus expect *petal width*, used in the tree root, to be the globally most relevant feature. When distinguishing between *versicolor* and *virginica* instead, the model relies heavily on *petal length* as exemplified in Figure 3. Finally, when separating between *setosa* and the two classes to the right of the root, we foresee *petal length* to have low importance, as it cannot be used by the model to classify a sample as belonging to the *setosa* class.

---

[1]Code available at `https://anonymous.4open.science/r/ICFI-D11F`

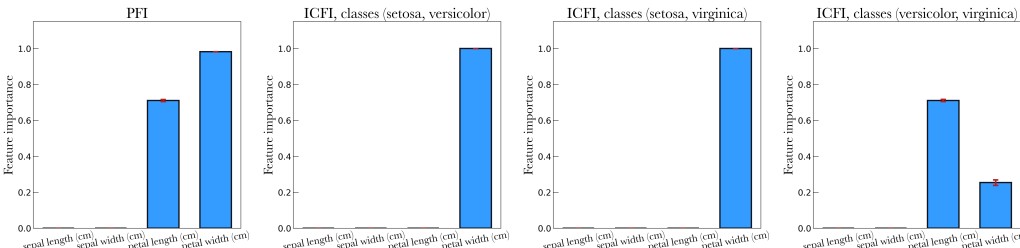

Figure 4: Global feature importance and ICFI for the model in Figure 3, fit on the *Iris* dataset. Global PFI (left) highlights the two features used by the model. ICFI computed between *setosa* and *versicolor* (right) and between *setosa* and *virginica* (center right), show how the model relies on the feature at the tree's root (*petal width*). Accordingly to the structure in Figure 3, ICFI between *versicolor* and *virginica* ranks *petal length* above *petal width*. Error bars represent the $95\%$ confidence interval. To aid an easier comparison between plots, the $y$ axis is standardized to the same range.

Figure 4 shows the global PFI feature importance (left), the other three plots display ICFI for all three class combinations. Error bars represent the $95\%$ confidence interval estimated through Bayesian inference exploiting Markov Chains Monte Carlo (MCMC); the strategy employed to compute confidence intervals in Figure 4, as well as in the rest of the paper, is outlined in Appendix A.1.

ICFI's rankings confirm the intuitions deriving from the tree structure. First of all, *sepal width* and *sepal length*'s importance is negligible as it should be. On a global level, the feature which is at the root of the tree, *petal width*, is also the top ranked. Thanks to ICFI we can instead see how the order is switched when the model tries to separate *versicolor* and *virginica*, in agreement with the structure showed in Figure 3. To separate *setosa* from the other two classes instead, the model mainly leverages *petal width*.

These traits of the model reasoning process, are not captured by classic global feature importance methods like, e.g., PFI, as they present generalized behavior and suffer from aggregation bias. Local methods instead fail to describe dynamics involving two entire classes, providing explanations for one particular flower sample.

## 4.2    4-CLASS LOGISTIC REGRESSION, TEXT

20 newsgroup (Mitchell, 1999) is a text dataset containing newsgroup posts on 20 topics. For the purpose of this experiment, background knowledge provided by using words as features, allows us to consider four classes: *Hockey*, *Baseball*, *IBM* and *Mac*, chosen as they are pairwise similar and difficult to separate. The four classes total 2635 samples in the training set and 1573 in the test set. We encode words as features, resulting in a dataset with 4525 dimensions. Further details on the data pre-processing strategy are provided in Appendix A.2. The model is a binary logistic regression model fitted for each label. Figure 5 displays the model's confusion matrix computed on test data.

The tiles highlighting the most misclassifications are the ones involving the *Hockey-Baseball* and the *IBM-Mac* combinations, which are the pairings involving the most similar classes and thus, the most difficult to separate. The model performing better for certain class combinations than in others, raises the question of which are the important features for each pair.

Figure 6 shows PFI, $ICFI^{Baseball\text{-}Hockey}$ and $ICFI^{IBM\text{-}Mac}$, displaying the top ten ranked features. Global feature importance, as expected, highlights features relevant in both tasks, e.g., *mac*, *apple*, *hockey* and *baseball*, summing-up the whole model behavior. In both ICFI plots showed in Figure 6, we can instead see how words related to the inspected classes are the most important ones. For example *mac* and *pc* are important features for the *Mac-IBM* class combination, while *nhl* and *pitcher* are within the highlighted features for *Baseball-Hockey*. Furthermore, the top spots are taken by features having high discriminant power between the classes of interest. Looking at $ICFI^{Baseball\text{-}Hockey}$ the first two ranked features are indeed *baseball* and *hockey*.

Furthermore, note that the feature importance values are relatively low w.r.t. the $[0, 1]$ range. This makes intuitively sense as the model has a high number of features to rely on. A measure with no

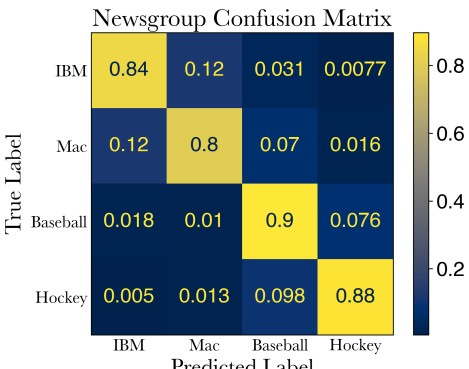

Figure 5: Newsgroup dataset normalized confusion matrix. The tiles showing the most misclassifications are the ones involving the *Hockey-Baseball* and the *IBM-Mac* pairs.

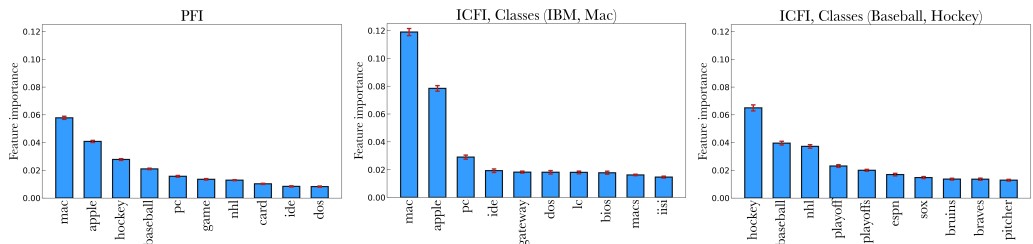

Figure 6: From left to right, PFI, ICFI of *IBM* and *Mac* and ICFI of *Hockey* and *Baseball*. The top ten ranked features are displayed. For the sake of comparison, plots are displayed with the same $y$ range. Error bars represent the $95\%$ confidence interval.

upper bound couldn't have lead to such consideration, as we would not have any reference value to compare the computed FI with. ICFI correctly retrieves features used to separate an arbitrary pair of classes: a level of insight lost due to aggregation bias in global FI metods. We can indeed notice how by looking at PFI's ranking, without background knowledge, we wouldn't be able to grasp which features are highly discriminative for which pair of classes.

### 4.3 COMPLEX MULTI-CLASS NEURAL NETWORK

We now consider model retraining to test the quality of ICFI's feature importance ranking. Conceptually, the better the feature ranking, the better a model trained with only the top $k$ most important features will perform. As ICFI computes a ranking for a pair of classes for the model to explain, the retraining is carried out using a *One versus One* strategy. A binary classification model is created for each class pair and fitted using ICFI's top $k$ features for each respective class pair. Each point is classified for each model and a final classification is obtained through a majority vote (Bishop, 2006).

We utilize four real-world multi-class classification datasets: *Dry Bean* (mis, 2020), *Penguins* (LTER & Gorman, 2016), *Vehicle silhouettes* (Mowforth & Shepherd), and *Wine* (Aeberhard & Forina, 1991), which have 13611, 342, 423, and 178 samples respectively. *Dry Bean* is a classification dataset of grains belonging to 7 different varieties of dry beans. Each record has 16 numerical features describing the grain's shape and dimension. The *Penguins* dataset contains 4 numerical features about three different species of penguins. The goal of *Vehicle silhouttes* is to classify a given silhouette, leveraging 18 numerical features, as one of four types of vehicles. Lastly, *Wine* leverages the quantities of 13 wine constituents to label each record as belonging to three different cultivars.

The model to explain is a feed-forward neural network, i.e., a black box model, which is trained for the multi-class classification problem at hand. ICFI is computed for each class pair and the top

$k$ features for different values of $k$ are selected. For each value of $k$, a *One vs One* classifier is then fitted, i.e., a neural network is trained for each class pair, with the final decision obtained by aggregating models' output through a majority vote. Each neural network, binary classifying a class pair, will use the top $k$ features highlighted as most important by ICFI computed for that class pair. The better the features, the higher the performance of the retrained *One vs One* (Borisov et al., 2019; Huang et al., 2020). We remark how the *One vs One* strategy is leveraged only in the evaluation step to assess ICFI's explanation quality. ICFI does not need a *One vs One* model for its computation and is, on the contrary, completely model agnostic.

We compare ICFI's rankings quality with other four feature selection strategies. We indeed compute PFI, global SHAP and global LIME feature importance on the neural network we seek to explain, choosing the top $k$ features for model retraining. In this scenario, each model in the *One vs One* classifier uses the same top $k$ features, as a global measure is used. In the fourth benchmark, features are selected randomly for each class pair. We include global methods in our comparison because of the absence of XAI methods natively having the same ICFI objective, and to show how the features important globally are not the most discriminative for each pair of classes. Three additional benchmarking strategies, where *state-of-the-art* global methods explain multiple binary models, one for each pair of classes, are discussed in Appendix A.3.

Figure 7 shows test accuracy at different $k$ values. Retraining based on ICFI consistently outperforms retraining leveraging PFI, global SHAP, global LIME and when features are chosen randomly. Specifically, the increase in performance is most evident when a low number of features is used, which is the most challenging setting and thus were the quality of the features' ranking matter the most. This indicates how features chosen with ICFI for each *One vs One* model carry more discriminative power than the features selected by global XAI methods. ICFI can thus effectively find features with high discriminative power to separate two classes, not suffering from aggregation bias.

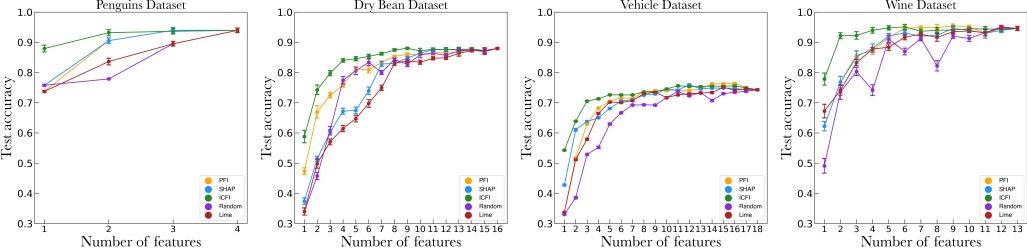

Figure 7: Test accuracy at different number of features selected. From left to right *Penguins*, *Dry Bean*, *Vehicle silhouettes* and *Wine* dataset. For the sake of better comparison the $y$ axis has the same range across all plots. Error bars show the $95\%$ confidence interval.

## 5 CONCLUSIONS

ICFI leads to more insights in a multi-class classification scenario, where current methods either suffer from aggregation bias or are tailored toward binary settings. ICFI quantifies feature importance in discriminating arbitrary pairs of classes. ICFI is a post-hoc, model-agnostic XAI method applicable to any existing ML model.

ICFI relies on merging the two inspected classes and measuring the performance improvement to measure the model's performance in separating the pair of classes. We use permutation to mimic the absence of a feature, allowing us to quantify its importance. ICFI's output is bounded to a well-defined range to make it more interpretable to human stakeholders. Evaluation has, through experiments, demonstrated ICFI's usefulness and relevance. We used ground-truth provided by an interpretable model and words in an NLP dataset to show how ICFI correctly retrieves features with high discriminative power for a pair of classes. Global methods on the same tasks fail to retrieve these insights, representing average behavior. Model retraining was used to display the quality of ICFI's feature ranking.

Future work could further explore different permutation strategies and feature attribution scores, thereby leveraging our idea for a diverse set of XAI scores.

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

## A  APPENDIX

### A.1  CONFIDENCE INTERVALS COMPUTATION

In each ICFI and PFI computation in the paper, the same strategy is employed to compute error bars, which relies on Bayesian inference. We run the feature importance computation 100 times where each run differs because of the randomization in the permutation procedure. As an approximation, we assume the process is modelled by a Gaussian likelihood,

$$P(x|\mu, \sigma) = \frac{1}{2\pi\sigma^2} \cdot e^{-\frac{(x-\mu)^2}{2\sigma^2}} \tag{5}$$

We infer its mean $\mu$ and standard deviation $\sigma$ through a Bayesian approach. We make a conservative choice for both priors using a uniform distribution defined in the interval $[0, 1]$. We sample the posterior using a Markov Chain Monte Carlo (MacKay, 2003), which allows us to skip the evidence computation.

We generate chains using Python's *emcee* package (Foreman-Mackey et al., 2013). For each run we generate 20 chains with 6000 samples each, using the first 1000 as burn-in. The sampled points from each chain are then merged together. For our purpose, we consider only the $\mu$ parameter samples, quantifying feature importance with its mean and identifying the 95% confidence interval excluding the first and last 2.5 percentile of the distribution.

Figure 8 showcases one MCMC chain without burn-in (left) and the resulting $\mu$ distribution (right), for the *petal length* feature in the top left plot in Figure 8; i.e. for the PFI computation of the *petal length* feature in the *Iris* dataset.

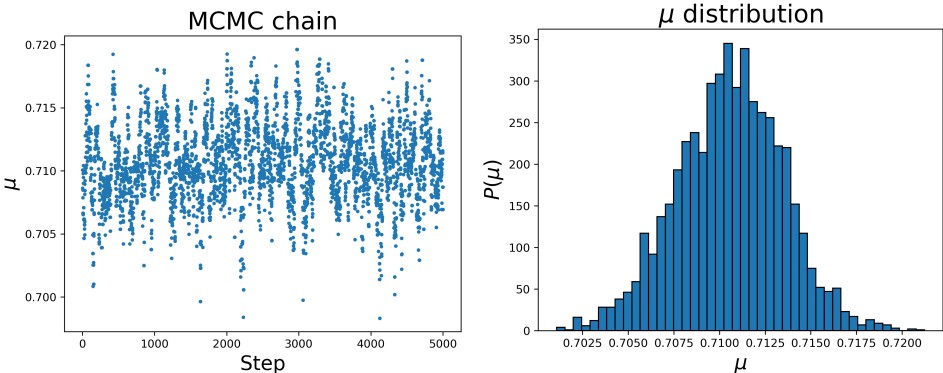

Figure 8: MCMC chain (left) and posterior marginal distribution of the $\mu$ parameter (right). The chain is run on the data obtained running 100 times the PFI algorithm on the *petal length* feature of the *Iris* dataset.

### A.2  20 NEWSGROUP PREPROCESSING

The data is preprocessed by creating a matrix representation of words count. A TD-IDF scheme (Baeza-Yates et al.) is then applied scaling down the impact of frequent tokens. Moreover, words appearing in more than half of the documents, or less than five times in total, are removed. This strategy allows us to rely on words as features, giving high interpretability. Crafting features with text embeddings, would instead imply features carrying less interpretability (Ribeiro et al., 2016).

### A.3  MODEL RETRAINING

In Section 4.3, we benchmarked ICFI with global methods in order to assess how aggregation bias hides discriminant features for a pair of classes. Global methods do not have the same ICFI's objective because they explain the model globally without accounting for inter-class relationships. Here,

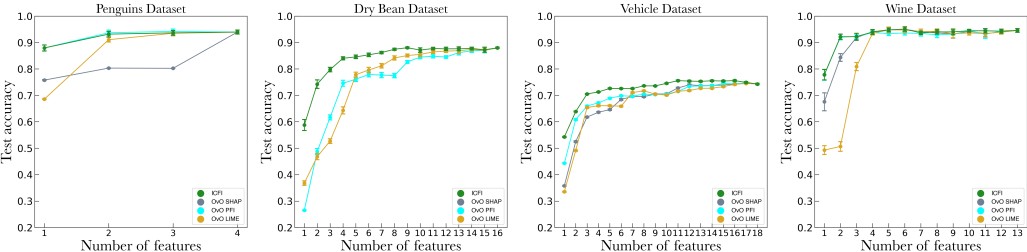

Figure 9: Test accuracy at different number of features selected. From left to right *Penguins*, *Dry Bean*, *Vehicle silhouettes* and *Wine* dataset. For the sake of better comparison the $y$ axis has the same range across all plots. Error bars show the $95\%$ confidence interval. OvO in the legend signals that the method has been used on each binary model of a *One vs One* approach.

in order to build benchmarks having the same objective as ICFI, we use the following strategy. Instead of fitting and explaining a single neural network, we fit and explain multiple binary models, one for each class combination. A feature ranking is retrieved for each pair of classes, using global SHAP, global LIME and PFI respectively on each binary model. Retraining is performed analogously to the benchmarks in Section 4.3.

For a fair comparison with ICFI explanations already produced in Section 4.3, we use a comparable total number of parameters w.r.t. the neural network explained in Section 4.3. We note that while ICFI is completely model agnostic and handles multi-classification natively, state-of-the-art FI methods require, to find discriminative features for a pair of classes, a binary model for each class combination (i.e. a *One vs One* approach). This is computationally demanding and sacrifices accuracy on the original task, on top of heavily constraining the classification strategy, compared to our earlier experiments.

Results are displayed in Figure 9, *OvO* in the legend signals that the XAI explanation has been computed on each binary model. ICFI is on par with PFI in the *Penguins* and *Wine* dataset while otherwise outperforming competing methods. ICFI achieves this without constraining in any way the classification strategy, not requiring a *One vs One* approach, demonstrating its novel contribution in multi-class classification.

### A.4 ICFI PROPERTIES

The purpose of this section is to show that ICFI's definition in Eq. 4 implies a non-negative feature importance quantification which is bounded between $0$ and $1$ and a measure which is symmetric on the pair of classes. Note that boundedness between $0$ and $1$, implies non-negativity. We will then prove:

1. $ICFI_j^{\sigma\rho} \geq 0$ .

2. $ICFI_j^{\sigma\rho} \leq 1$ .

3. $ICFI_j^{\sigma\rho} = ICFI_j^{\rho\sigma}$ .

*Proof.*

1. We need to show that

$$\frac{\left|\Delta\tilde{\mathcal{R}}_j^{\sigma\rho} - \Delta\mathcal{R}^{\sigma\rho}\right|}{\Delta\mathcal{R}^{\sigma\rho}} \geq 0 \iff \Delta\mathcal{R}^{\sigma\rho} > 0 \quad . \tag{6}$$

It is left to prove that $\Delta R^{\sigma\rho} = \mathcal{R} - \mathcal{R}^{\sigma\rho} > 0$. If the relation is true for every single sample, it will consequently stay true when taking the average. The proof is shown for the cross-entropy loss: the scenario adopted in this paper's experiments and the most used loss in multi-class classification.

There are two possibilities to take into account:

- If the true label is neither $\sigma$ or $\rho$, $\tilde{\mathcal{R}}_j$ and $\mathcal{R}_j^{\tilde{\sigma}\rho}$ have the same value. The two original probabilities and the merged one are indeed multiplied by $0$ in the cross-entropy formulation.
- Without loss of generality, taking $\sigma$ as the true class, the cross entropy contribution for $\mathcal{R}$ is $-log\ p_\sigma$. With $p_\sigma$ the model probability for class $\sigma$. The merged one is instead $-log\ (p_\sigma + p_\rho)$ with the difference being

$$log\ (p_\sigma + p_\rho) - log\ p_\sigma =$$

$$= log\ \frac{p_\sigma + p_\rho}{p_\sigma} = log\ \left(1 + \frac{p_\rho}{p_\sigma}\right) > 0 \quad . \tag{7}$$

Note that in eq. 7 and in the cross-entropy computation, the probabilities are clipped avoiding $p_\rho$ and $p_\sigma$ to be exactly $0$.

2. To show that Eq. 4 always evaluates $\leq 1$ we need to show that

$$-\frac{1}{1 + \left|\Delta\tilde{\mathcal{R}}_j^{\sigma\rho} - \Delta R^{\sigma\rho}\right|/\Delta R^{\sigma\rho}} \leq 0 \quad , \tag{8}$$

which is immediate from Eq. 6.

3. The merge operation is completely symmetric and there are no operational differences in computing $ICFI^{\sigma\rho}$ and $ICFI^{\rho\sigma}$.

$\square$

