# OpenReview forum: "ICFI: a Feature Importance Measure For Multi-Class Classification"
_ICLR.cc/2025/Conference — Submitted to ICLR 2025_

### Official Review · Reviewer_U1Uf · 2024-10-27

**Soundness:** 2
**Presentation:** 2
**Contribution:** 2
**Rating:** 5
**Confidence:** 4

**Summary:**

The paper introduces Inter-Class Feature Importance (ICFI), a model agnostic method to identify important features to distinguish between a pair of classes for multi-class classification tasks. ICFI is a permutation based feature selection method that measures the drop/increase in model performance when merging a pair of classes. Intuitively if permutations in values of a feature results in bigger model performance deterioration when merging two classes, the feature plays a more important role in distinguishing between those two classes.
Authors study the feature importance from ICFI for various class pairs for a couple of commonly used data sets (Iris and 20 news group) and compare them against a global method (i.e. Permutation Feature Importance). They also, study test accuracy based on number of features used in training a multi-class classifier over four data sets when features are selected based on ICFI and three alternative methods, i.e. PFI, Shapley additive explanations (SHAP), and a random feature selector and show that ICFI outperforms other methods in their studies.

**Strengths:**

- The paper has done a decent job covering related literature.
- The idea of merging classes, permuting a feature and measuring the delta in empirical error makes sense and is well explained.
- The XAI problem and in particular identifying important features for multi-class classification tasks by looking into features that can help distinguishing between a given pair of classes is a worthwhile problem to solve witch wide use cases.

**Weaknesses:**

- The biggest weakness of the method is the lack of clarity in the proposed method. It is not clear how the top important features identified by ICFI for various pairs of classes are combined to train a multi-class classifier. For example, in Fig. 7, test accuracy is measured at different numbers of selected features and it is not explained how important features identified by ICFI across various pairs of classes are being selected.
- Along the same lines of the first point, it is not clear if ICFI is expected to provide explainability at global level or in terms of distinguishing between a pair of classes only. If the goal is the former, then authors should have discussed more details about how they combine feature importance of various pairs of classes and come up with a global ranking. If it is the latter, then it's not clear why they compare ICFI with global methods such as FPI (e.g. in Fig. 4).
- Some of the fundamental assumptions/claims of the paper needs further clarification. For example, in the abstract section, authors state that "In multi-class classification, current methods fail to explain inter-class relationships as they either provide explanations for binary classification only, or suffer from aggregation bias", however, there are methods such as LIME for which none of the listed limitations applies.
- Along the previous point, the alternative methods studied in the paper, do not include methods such as LIME that theoretically do not suffer from the stated limitations for existing methods.
- Lastly, the limitations of ICFI are not discussed in the paper. For example, how can ICFI capture the interactions between two or more features, as a concrete case consider a couple of features that do not show strong performance gain on their own while their combination could play a significant role in the predictive power of the model.

**Questions:**

- I can imagine that for the 20 news group data set, since features are words, permuting a feature means replacing it with an unused token/word in the vocabulary (e.g. NULL)? It would be useful if authors can clarify.
- Other questions are captured in the first two bullets of the weakness box.

---

> ### Author Response · Authors · 2024-11-17
>
> We would like to thank the reviewer for appreciating the problem in our paper as reasonable.
>
> - Weakness 1) ICFI is a method to explain an already trained machine learning model. After computing ICFI there is no need for further training. In Figure 7 we select the most important k features (for different values of k) for each pair of classes. Each feature selection is then leveraged when training the binary models composing the One vs One evaluation strategy. We want to clarify (here and in the revised version of the paper) that One vs One is only used to evaluate the explanations, but is not part of the ICFI computation as such.
> - Weakness 2) ICFI is not a global method in the traditional sense, and indeed outputs feature importance for pairs of classes specifically. As ICFI is the first method that provides such class pair specific explanations, there are no direct competitors. To still be able to assess its benefit, we compare with state-of-the-art global methods and random feature selection as a baseline.
> For evaluation, we indeed use global feature ranking to choose features for each binary model used in the model retraining experiments. Contrary to ICFI, due to their global nature, PFI, SHAP and LIME (added in the new revised version of the paper) select the same features for each pair of classes (and thus for each binary model). We show that such global method explanations do not effectively capture discriminative features for pairs of classes. This confirms that their global ranking suffers from aggregation bias, and insights at class level granularity are lost due to averaging.
> - Weakness 3) LIME’s global explanations aggregate explanations over all instances, yielding a single ranking. As the experiments confirm, this averaging suffers from aggregation bias as features important for discrimination between pairs of classes as identified by ICFI are missed in this global view. Consider a feature highly relevant to discriminate between two classes only, in the global ranking which takes into account every class, its importance may be hidden by features important for a greater number of instances.
> - Weakness 4) We have expanded the benchmarking in Section 4.3 by adding comparison with LIME. As for other global methods, the LIME results show that it fails to capture feature importance for pairs of classes, and is clearly outperformed by our ICFI method.
> - Weakness 5) Our proposed ICFI method considers single features and computes a feature attribution for each feature individually, as also done by state-of-the-art feature importance methods like LIME. Considering feature interaction is an interesting idea that applies to feature importance measures like PFI and SHAP. Considering different permutation strategies, hinted in the feature works, could also try to tackle the feature interaction aspect.
> - Question 1) For the textual data, we use the well known TF-IDF approach to represent for each word its term frequency (how often it appears in a document) weighted by inverse document frequency (how rare the word is across the document collection). TF-IDF values are thus the features' numerical values, and permutation is performed as in the other tabular data sets, i.e., we select one of the N! feature permutations at random.

---

> ### Comment · Reviewer_U1Uf · 2024-11-24
> **Thank you for your explanations**
>
> I'd like to thank the authors for their detailed explanations and response to question 1.
> Following on your explanations in the second bullet point from your comments, have the authors considered either of these two scenarios?
>
> 1- Compare ICFI with global methods for binary classification tasks?
>
> 2- Use ICFI under multi-class classification (same way as it is used in the paper) to select top features to train a new one vs. one classifier and then compare the performance of this model vs. a similar binary classifier model that is trained by using top features selected with global methods from an original binary classifier?
>
> Such experiments can shed light into how well ICFI can learn top features for a one vs. one classifier in a multi-class classification setting compared with the ideal scenario that requires more computational power.
>
> To summarize, after reading the authors' comments, I have no further questions related to the first, third, and fourth bullet in my comments under the weakness section. I also have no further questions about my question under the first bullet in the question section. **I would also like to change my overall rating from 3 to 5**.

---

> > ### Author Response · Authors · 2024-11-27
> >
> > We appreciate the reviewer's responses and continued interest in further improving our paper.
> > - We did not include a comparison of ICFI for binary classification tasks because this corresponds to a traditional global setting. When we have a binary task, merging the only two classes will cause the empirical risk evaluated after merging to be zero. We thus considered only empirical evaluation for the most novel contribution, which is for multi-class classification tasks.
> > - We appreciate this suggestion for further in-depth benchmarking of ICFI. Following the reviewer's suggestion we expand the already present analysis in Section 4.3: the current benchmark fits and explains a single neural network, we complement the analysis by fitting multiple binary models (one for each class combination). To capture important features for discriminating a pair of classes we use state of the art global methods (LIME, PFI and SHAP) for each binary classifier. For a fair comparison with ICFI explanations already produced in Section 4.3, we use a comparable total number of parameters w.r.t. the neural network explained by ICFI.
> > We note that while ICFI is completely model agnostic and handles multi-classification natively, the new benchmarking strategy requires a One vs One approach, so one model is fitted for each class combination. This is computationally demanding and sacrifices accuracy on the original task, on top of heavily constraining the classification strategy (compared to our earlier experiments).
> > We added this benchmarking strategy in the latest revision of our paper (the main part of the discussion is in the appendix for space reasons). In this constrained setting, ICFI is on par with PFI in the Wine and Penguins dataset while otherwise outperforming competing methods. Thus, in order to output feature importance to separate a pair of classes, current state-of-the-art methods need to constrain the classification strategy to a One vs One approach. In this scenario ICFI (which instead is completely model agnostic) generally outperforms competing methods while allowing to build explanations on any fitted model. This, together with the already implemented benchmarks, demonstrates ICFI's novel contribution in the unrestricted multi-class classification task.

---

### Official Review · Reviewer_bFBc · 2024-10-29

**Soundness:** 2
**Presentation:** 2
**Contribution:** 2
**Rating:** 5
**Confidence:** 4

**Summary:**

This paper focuses on the problem that in the multi-class classification scenario, current methods fail to explain inter-class relationships as they either provide explanations for binary classification only or suffer from aggregation bias. In a multi-class classification scenario, features may carry discriminative power to separate some of the classes while being otherwise less relevant. Focusing on this problem, the authors proposed an Inter-Class Feature Importance (ICFI), to score the feature's importance between an arbitrary pair of classes.

**Strengths:**

This paper presents an "Inter-Class Feature Importance (ICFI)" method, specifically addressing the problem of inter-class feature importance in multi-class classification, filling the gap in the measurement of inter-class relationship in existing methods. The method can provide model-agnostic explanations without the need to train additional models, addressing the limitations of existing methods in measuring feature importance in multi-class scenario.

**Weaknesses:**

(1)The paper lacks enough innovation. ICFI avoids the aggregation bias commonly seen in traditional global feature importance methods by introducing a feature importance assessment based on class pairs and feature permuting. At present, some relevant studies have focused on feature permuting and class pairs, such as PFI, SHAP, and LOO. The ICFI method proposed in this paper mainly evaluates the importance of features through feature permuting, and its method mechanism is similar to PFI and SHAP. Therefore, although ICFI has some potential contribution to multi-class feature importance assessment, it is not innovative enough to clearly distinguish ICFI from existing methods.

(2) The writing of this paper should be further improved. There are many writing problems in the paper, which seriously affects the reading. The following are some of the problems I found:
--- On page 2, line 67, "a ML" should be "an ML". On page 2, line 83 has the same error.
--- "Fig 3", and "Figure 3" all appear in this paper. Ensure consistent figure references.
--- On page 3, line 152, "Fig. 8 shows the decision...". "Fig. 8" should be Fig. 1.
--- On page 4, line 205, the symbol "D" lacks definition.
--- On page 8, line 395, "Figure 8 shows the global PFI feature importance...", "Fig. 8" should be Fig. 4.
--- On page 10, in the Fig. 7, the "Shap" should be consistent with "SHAP" in this paper.
--- The description of the figure title is too complicated in this paper, so it is recommended to simplify the figure title to improve readability and add a serial number to the sub-graph for easy reference and understanding.

**Questions:**

In addition to the weaknesses I have already listed, I have the following questions:
(1)In Section 4.3, four different datasets were selected to verify the performance of ICFI in different tasks, but the complexity of these datasets is limited, and it is recommended to further extend the experiment to complex datasets with high dimensional and larger number of categories.

(2)The idea of assessing feature importance through feature permuting of ICFI is similar to that of PFI. Please analyze the uniqueness and advantages of ICFI similarly.

(3)In the Section 4.3, ICFI was compared to SHAP, PFI, and random strategies, why not choose LIME and PDP as the comparison algorithms?

(4)In this paper, the One-vs-One strategy is used to train the model and perform majority voting, but this method may incur high computational costs when the number of categories is high. It is suggested that the authors add an analysis of the computational efficiency of ICFI .

---

> ### Author Response · Authors · 2024-11-17
>
> We appreciate the reviewer comments, in particular noting that ICFI fills a gap in multi-class explanations.
>
> - Weakness 1 + Question 2) While ICFI uses permutation, as do PFI and SHAP, this is the only similarity (i.e., how we simulate the absence of the feature). Neither of the two methods PFI or global SHAP, for example, merges classes nor takes into account any inter-class relationship. PFI and SHAP average contributions from all the instances to provide global explanations across classes, risking to overlook features discriminative only for a single pair of classes  (aggregation bias). SHAP local explanations fail to capture behaviours involving multiple instances.  Moreover, SHAP can output explanations only toward one class, meaning for multi-class analysis one needs to resort to a One vs All approach.
> - Weakness 2) We thank the reviewer for the detailed comments, which we have fixed in the revised paper version.
> - Question 1) The datasets used in Section 4.3 with a relatively low dimensionality have been chosen in order to allow visual analysis, but analyse high dimensional data in Section 4.2, with more than 4000 features. We indeed retrain using a One vs One strategy, increasing the number of features selected by one at each step, more than 4000 features hinder the ease of this visualisation. Analysing more than 4000 features (in section 4.2) shows that ICFI performs on high dimensional data.
> - Question 3) PDP does not output a feature importance ranking and is thus not suited for the ranking evaluation in Section 4.3
> In Section 4.3 we indeed perform feature selection for every binary model of the One vs One strategy (which are trained only for evaluation purposes, and in a real application are not needed for ICFI), requiring thus a feature ranking to select the most important ones.
> For completeness, we add retraining using LIME's ranking in the revised version of our paper. As the results show, ICFI clearly outperforms LIME.
> - Question 4) We further clarify in the revised version of our paper that the One vs One strategy is purely an evaluation methodology, but not needed to compute ICFI. ICFI is completely model agnostic and can thus be computed on top of any black box machine learning model. The One vs One evaluation strategy is employed in Section 4.3 only to evaluate the quality of ICFI’s feature ranking and to benchmark it against other rankings.  This evaluation strategy is the following: train a black box model, compute ICFI for each class combination. Compute global SHAP, PFI, LIME (the latter added in the newest version of the paper), and a ranking based on choosing features at random. ICFI and the random strategy are thus the only two methods outputting a different ranking for every class pair while the other global methods output only one ranking. For evaluation, we then train a binary model for each pair of classes, aggregating predictions using majority voting (One vs One). This is done for the k most important features, for different values of k.
> Retraining based on ICFI’s ranking thus implies selecting (potentially) different features for every class combination while global method always return the same feature choices (for all class pairs). In the experiments, we show that the most important features globally are not necessarily the most important for discriminating between a particular pair of classes. Global methods show lower performance than ICFI, because averaging causes insights to be lost due to aggregation bias.

---

### Official Review · Reviewer_H9xb · 2024-11-03

**Soundness:** 3
**Presentation:** 3
**Contribution:** 2
**Rating:** 5
**Confidence:** 3

**Summary:**

This submission proposes Inter-Class Feature Importance (ICFI) measure, which scores the feature importance to discriminate between pairs of classes in multi-class classification (MCC). The measure is formulated in Section 3.1, and experiments are conducted in Section 4 to assess the potential usefulness of the proposed measure.

The notion of ICFI, which is compactly described in Eq (4), is interesting (at least from the conceptual level). However, it seems that the scalability with respect to both the number of features and classes is a crucial challenge in practice.

**Strengths:**

S1: I think the notion of ICFI is intuitive and interesting.

S2: Empirical results seem to be in favor of the proposed ICFI measure.

**Weaknesses:**

W1:  It seems that the scalability with respect to both the number of features and classes is a crucial challenge in practice.

W2:  It might be beneficial to discuss how ICFI can be adapted to enlarge the existing set of instance-level feature importance measures.

**Questions:**

Q/S1: For each pair of classes, for each feature of interest, did you train one model (after permuting the corresponding feature of interest)?

Q/S2:  Could you comment on the scalability of the proposed ICFI measure? How many features and classes one might handle by using the proposed ICFI measure? Can it be scalable up to image data sets?

Q/S3: Could you discuss how ICFI can be adapted to enlarge the existing set of instance-level feature importance measures?

---

> ### Author Response · Authors · 2024-11-17
>
> We thank the reviewer for the feedback and for appreciating that ICFI is intuitive and interesting.
>
> - Question 1) In order to compute ICFI we do not train any additional model. After the model to explain has been trained, ICFI is computed for each feature and for any selected pair of classes. After feature permutation, ICFI requires only evaluations of the original (black box) model and no further training. Only in Section 4.3, we use a retraining strategy to further evaluate the quality of explanations.
> Model retraining is used only to evaluate the already produced ICFI explanations here, and is performed when ICFI is already computed, exploring its and the other benchmarked methods’ ranking.
> We will make this more explicit in a revised version of our paper.
> - Question 2) Regarding scalability, ICFI averages the contribution of each instance and thus has a linear dependency on the number of data points. The computation is independent for each feature so a complete ranking on every feature scales linearly also with the number of features (and could be easily parallelized). We would like to point out that this is the same computational complexity as that of PFI. ICFI is computationally much cheaper than SHAP, as we do not need to evaluate every feature coalition but practically just one - for the permutation. To output a global explanation, LIME is computed for each instance and its complexity also depends on the surrogate interpretable model fitted for each single instance.
> ICFI can also be used on image data, as also done with SHAP; permutation would then be done on more complex features than single pixels, which would likely not affect classification much. E.g. super-pixels are commonly used with SHAP, and ICFI could use them in the same manner. ICFI is computationally much cheaper than SHAP and its application on image data would thus also benefit from a speedier computation compared to SHAP.
> - Question 3) ICFI is not a local (instance-based) method and explains the reasoning process of the model regarding two entire classes, finding important features in the task of separating the feature pair.
> We thank the reviewer for suggesting an instance-level extension of ICFI which could be implemented repeating the permutation we already do for a single instance, with multiple selected permuted samples.

---

### Author Response · Authors · 2024-11-17

We thank the reviewers for the detailed feedback provided and we appreciate that ICFI is seen as intuitive, interesting, filling a gap in the literature with a reasonable and well-explained merging idea.
We appreciate the comments for further improving our paper. In particular, we revise our description of the evaluation using model re-training, leveraging a One vs One strategy, to emphasise that is an assessment strategy only.

The One vs One strategy is not utilised as part of computing ICFI. Our proposed measure is model-agnostic and can, therefore, be applied on top of any black box model and does not need to be able to calculate a One vs One strategy, with the limitations this would have as rightfully pointed out by the reviewers.
In Section 4.3, the One vs One strategy is leveraged to assess the quality of feature rankings and to benchmark ICFI against other ranking methods. Because of the lack of competing methods, we demonstrate the benefit of our novel ICFI method by benchmarking against global methods and a baseline with a random strategy.
Our evaluation approach in Section 4.3 starts by training a black-box model. We then compute global SHAP, PFI, LIME (introduced in the revised version of the paper) along with ICFI and a random feature selection for each class combination. Among these, ICFI and the random strategy are the only methods that produce different rankings for each class pair, whereas global methods provide a single ranking across all class pairs.
We continue by training a binary classifier for each pair of classes, aggregating decisions through majority voting (One vs One). This process involves selecting only the top k features, considering multiple values for k. When retraining based on ICFI rankings, different features for each class pair are selected, whereas global methods use the same feature selection, as they lack class-pair-specific rankings.
Our experiments demonstrate that the features deemed most important globally are not necessarily the most important for distinguishing between specific pairs of classes. Feature selection driven by global methods results in lower performance compared to ICFI, as the aggregation inherent in global approaches obscures valuable insights about individual class pairs, leading to aggregation bias.

---

### Meta-Review · Area_Chair_ThsT · 2024-12-21

**Metareview:**

The paper introduces a new measure for feature importance suited for multi-classification problems. The measure focuses on intra-class relationships by scoring features to discriminate between an arbitrary pair of classes. This behavior is ignored by current state-of-the-art approaches.

The Reviewers state that the method is interesting, intuitive, and important in the context of explainable artificial intelligence. They underline that the method is model-agnostic. The empirical results are promising. The Authors extensively present the related work. Nevertheless, the Reviewers remarked that the paper is not entire clear, for example, the method, its usage and its purpose. The initial version of the paper was lacking a comparison to an important related method.

Overall, the contribution seems to be interesting, but not good enough for a top ML conference. I encourage the Authors to improve clarify of the paper to avoid any confusions around the method and the experimental studies/use cases.

**Additional Comments On Reviewer Discussion:**

The Authors were able to clarify many doubts of the Reviewers. They also conducted additional experiments. This resulted in improved ratings, but not above the acceptance bar.

---

### Decision · Program_Chairs · 2025-01-22

Reject